# The magnesium transporter A is activated by cardiolipin and is highly sensitive to free magnesium in vitro

Saranya Subramani[1], Harmonie Perdreau-Dahl[1,2], Jens Preben Morth[1,2]*

[1]Norwegian Centre of Molecular Medicine, Nordic EMBL Partnership University of Oslo, Oslo, Norway; [2]Institute for Experimental Medical Research, Oslo University Hospital, Oslo, Norway

**Abstract** The magnesium transporter A (MgtA) is a specialized P-type ATPase, believed to import $Mg^{2+}$ into the cytoplasm. In *Salmonella typhimurium* and *Escherichia coli*, the virulence determining two-component system PhoQ/PhoP regulates the transcription of *mgtA* gene by sensing $Mg^{2+}$ concentrations in the periplasm. However, the factors that affect MgtA function are not known. This study demonstrates, for the first time, that MgtA is highly dependent on anionic phospholipids and in particular, cardiolipin. Colocalization studies confirm that MgtA is found in the cardiolipin lipid domains in the membrane. The head group of cardiolipin plays major role in activation of MgtA suggesting that cardiolipin may act as a $Mg^{2+}$ chaperone for MgtA. We further show that MgtA is highly sensitive to free $Mg^{2+}$ ($Mg^{2+}_{free}$) levels in the solution. MgtA is activated when the $Mg^{2+}_{free}$ concentration is reduced below 10 μM and is strongly inhibited above 1 mM, indicating that $Mg^{2+}_{free}$ acts as product inhibitor. Combined, our findings conclude that MgtA may act as a sensor as well as a transporter of $Mg^{2+}$.

*For correspondence: j.p.morth@ncmm.uio.no

**Competing interests:** The authors declare that no competing interests exist.

## Introduction

Magnesium is the most abundant divalent cation in biological systems and is an essential requirement for all living cells (*Reinhart, 1988*). $Mg^{2+}$ has diverse biological roles, ranging from being an essential cofactor in ATP-mediated enzymatic reactions to being a signaling molecule that activates important virulence systems in bacteria (*Groisman et al., 2013*). $Mg^{2+}$ homeostasis is well studied in Gram-negative bacteria like *S. typhimurium* and *E. coli* (*Papp-Wallace and Maguire, 2008*). Three classes of $Mg^{2+}$ transporters have been identified in bacteria: CorA, MgtE and MgtA (magnesium transporter A)/MgtB. Based on the $Mg^{2+}$ transport studies, Snavely et al., proposed that CorA transports $Mg^{2+}$ under normal $Mg^{2+}$ levels, whereas MgtA and MgtB transport $Mg^{2+}$ when bacteria faces low $Mg^{2+}$ condition (*Snavely et al., 1991*). Later Garcia Vescovi et al., identified that the low $Mg^{2+}$ levels in the periplasm activate the PhoQ/P system (*Véscovi et al., 1996*), which in turn induces the expression of genes essential for adapting the $Mg^{2+}$ limiting environments (*Monsieurs et al., 2005*). One of the genes was found to be *mgtA*. In addition to PhoQ/P mediated activation, a $Mg^{2+}$ sensing riboswitch at the 5' leader region was found to activate the transcription of *mgtA* gene when the $Mg^{2+}$ level in cytoplasm falls below a certain threshold (*Cromie et al., 2006*). Therefore, both the intracellular and extracellular $Mg^{2+}$ concentrations regulate transcription of the *mgtA* gene. Upon translation, MgtA is believed to transport $Mg^{2+}$ from the periplasm into the cytoplasm under $Mg^{2+}$ depriving conditions (*Snavely et al., 1989*). It has been shown that deletion of the *mgtA* gene affects the survival of *S. typhimurium* at higher temperatures and also promotes lysis in *Streptococcus pneumonia* (*O'Connor et al., 2009*; *Neef et al., 2011*) .

**eLife digest** Magnesium is an essential element for living cells, meaning that organisms from bacteria to humans need magnesium to survive. All cells are surrounded by a membrane made of fatty molecules called lipids, which is also embedded with proteins. Magnesium, like other metal ions, is transported inside cells across the cell's membrane by specific membrane proteins.

A species of gut bacteria called *E. coli* has two separate magnesium transport systems: one that works at high concentrations of magnesium and one at lower concentrations. The latter system involves a membrane protein called magnesium transporter A (or MgtA for short), which works like a molecular pump. However, it was not known exactly how this transporter was affected by magnesium nor how sensitive it was to this divalent metal ion. It was also unclear whether MgtA worked alone in the bacterial membrane or if it worked in conjunction with other molecules.

Now Subramani et al. have managed to show that MgtA can sense magnesium ions down to micromolar concentrations, which is the equivalent to a pinch (1 gram) of magnesium salt in 10,000 liters of water. The experiments also showed that this detection system depended on a specific lipid molecule in the membrane called cardiolipin. MgtA and cardiolipin were found together in the membrane of living *E. coli* suggesting that the two do indeed work together.

The discovery that a membrane transporter that pumps ions needs cardiolipin to work suggests that cells could indirectly control the movement of ions by changing the levels of specific lipids in their membranes. Subramani et al. now hope to use techniques, such as X-ray crystallography, to visualize how magnesium and cardiolipin bind to MtgA and explore how the three molecules work together as a complete system. Information about these interactions could in the future help researchers understand how these bacteria try to protect themself in the hostile environment in the human gut or cells of the immune systems. Further studies of this system could be used to develop biological sensors for magnesium or to design antibiotics that interfere with the magnesium transporter to treat bacterial infections.

MgtA belongs to the P3 subfamily of P-type ATPases (*Palmgren and Axelsen, 1998*). The P3 family is subdivided into P3A and P3B. The P3A family is dominated by $H^+$-ATPases found in plants (*Palmgren, 2001*), while P3B contains $Mg^{2+}$ ATPases, found to be dominant amongst prokaryotes (*Kühlbrandt, 2004*). Recently, a close homolog of MgtA was reported in *Petunia hybrida* (PH1), which lacks the ability to induce a proton gradient. PH1 was proposed to support the proton pumping performed by the P3A proton pump PH5, but evidence for magnesium transport was not established (*Faraco et al., 2014*). In general, MgtA is found in bacteria, archaea, fungi and plants (See also *Figure 1—figure supplement 1–2*). MgtA has ten predicted transmembrane helices (TM), a nucleotide binding domain (N), a phosphorylation domain (P) and an actuator domain (A) (*Figure 1A*). The Post-Albers cycle describes the transport mechanism of a P-type ATPase (*Albers, 1967*; *Post et al., 1969*). During the transport cycle, P-type ATPases alternate between the E1 and E2 states with different intermediate conformational states (*Palmgren and Nissen, 2011*). The E1 state has high affinity and is open for ions binding from the cytoplasm. Ion binding induces autophosphorylation by transferring the γ-phosphate from an ATP molecule to a conserved aspartate residue in the DKTGT consensus motif of the P domain forming the E1P state (*Lutsenko and Kaplan, 1995*). In *E. coli* MgtA (ecMgtA), the phosphorylated aspartate corresponds to Asp 373. Phosphorylation induces domain rearrangements leading to the E2P state, which is now open to the periplasmic side. E2P has low affinity for the bound ions and high affinity for the counter ions ($Mg^{2+}$ in case of MgtA). The exchange of counter ions dephosphorylates the enzyme and forms the E2 state. Further conformational changes return the enzyme back to the E1 state and the counterions are released into the cytoplasm, thus completing the transport cycle. A possible Post-Albers scheme with the four steps transport cycle can be visualized for MgtA and is given in *Figure 1B*. Since MgtA imports $Mg^{2+}$ into the cytoplasm, $Mg^{2+}$ is predicted to have high affinity to the E2P state. However, the stoichiometry and electrogenic nature of magnesium transport by MgtA remains unknown.

Over the past two decades, detailed studies of *mgtA* transcriptional regulation has provided a clear picture of the mechanisms through which *S. typhimurium* and *E. coli* tightly control *mgtA* gene

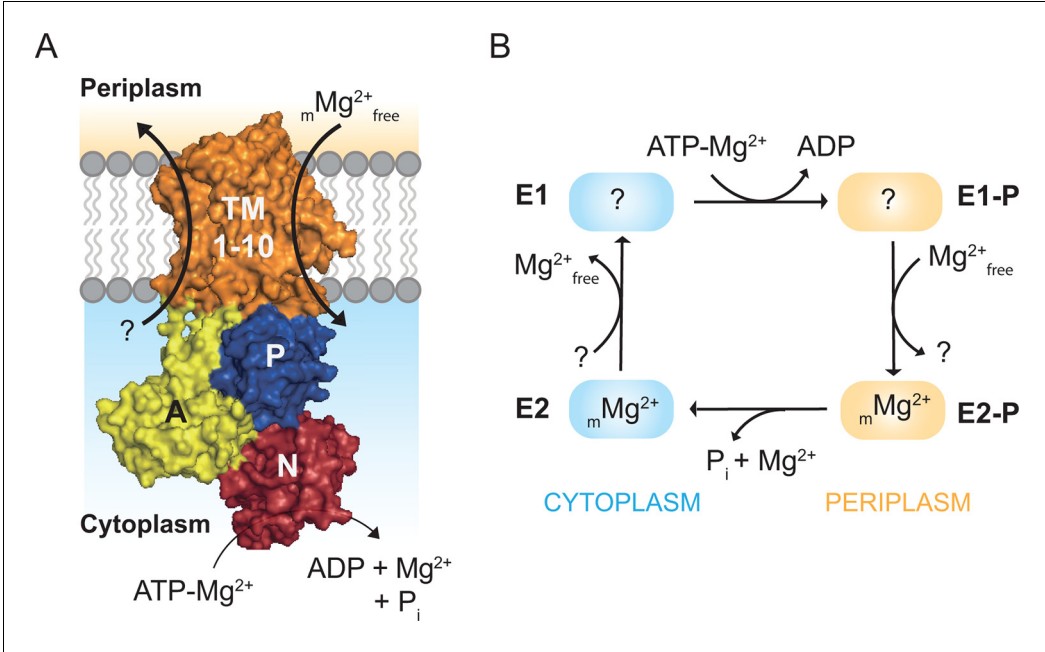

**Figure 1.** Magnesium uptake scheme for MgtA. (**A**) Cartoon representation of MgtA showing transmembrane domain (TM), actuator domain (A), phosphorylation domain (P) and nucleotide binding domain (N). MgtA transports $Mg^{2+}$ ions into cytoplasm as a function of ATP hydrolysis, (m) represent the unknown stoichiometry. (**B**) Proposed Post-Albers reaction scheme (**Albers, 1967**; **Post et al., 1969**) adapted for MgtA.

The following figure supplements are available for figure 1:

**Figure supplement 1.** Phylogenetic tree showing distribution of MgtA homologs among four kingdom of life: eubacteria (yellow), archaea (grey), fungi and slime mold (blue) and the plant kingdom (green).

**Figure supplement 2.** Multiple sequence alignment of selected MgtA homologs including the pig $Na^+/K^+$-ATPase (NKA1) and the human $H^+/K^+$-ATPase.

expression (**Groisman et al., 2013**). Despite these detailed transcriptional regulatory studies, very little is known about the biochemical mechanism or factors influencing the $Mg^{2+}$ transport by MgtA. Here, we present the first biochemical characterization of purified MgtA from *E. coli*. We show that anionic phospholipids, particularly cardiolipin (CL) are crucial for in vitro activation of MgtA. Further the overexpressed MgtA colocalizes with CL in vivo. We also show that MgtA ATPase activity is highly sensitive to $Mg^{2+}_{free}$ concentration and that $Mg^{2+}$ acts as an inhibitor at higher concentrations. Our findings reveal for the first time, the effect of anionic lipids and free $Mg^{2+}$ on MgtA function.

## Results

### Expression and purification of MgtA

In this study, MgtA was successfully overexpressed in its native host *E. coli* with yields up to 4 mg per liter of LB medium. The overexpression of MgtA is clearly visible as a band both in the cell lysate and membrane fraction as shown in SDS-PAGE gel (*Figure 2A*). Initial affinity purification with His-tag yielded >85% pure protein, additional size exclusion chromatography (SEC) increased the purity up to >95%, resulting in a highly purified protein for in vitro biochemical studies.

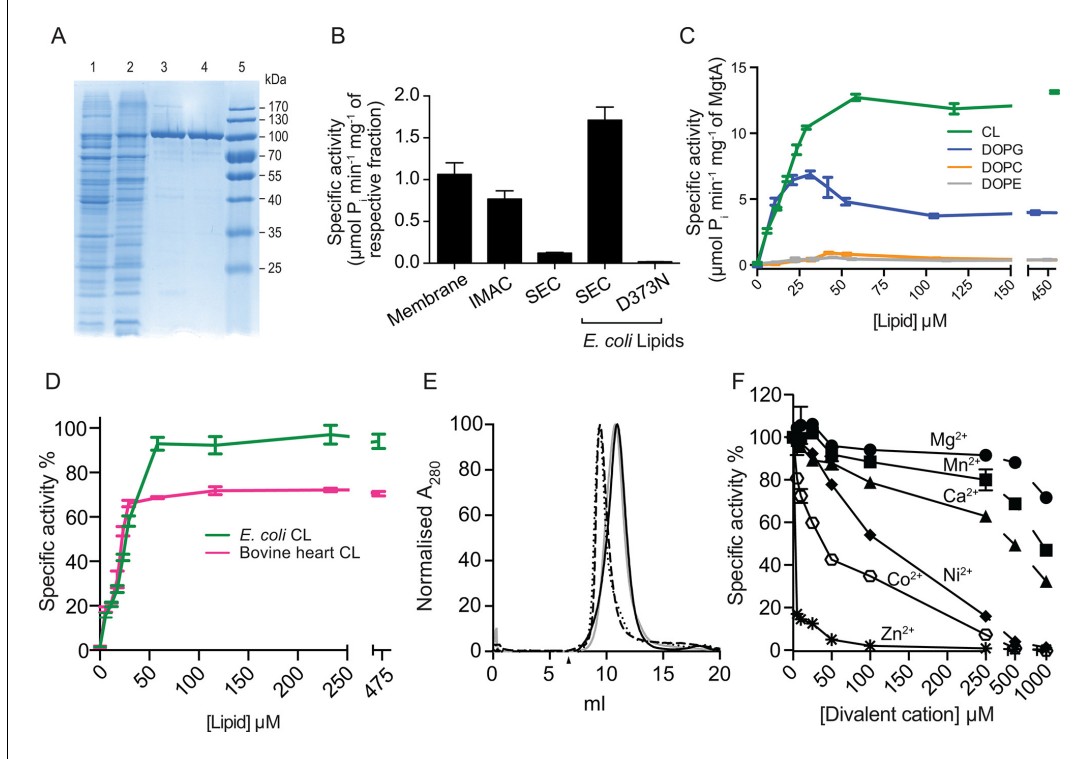

**Figure 2.** Biochemical characterization and cardiolipin dependency. (**A**) Samples taken throughout the purification process were subjected to SDS-PAGE. Lane 1- Cell lysate, Lane 2- Membrane fraction, Lane 3- $Ni^{2+}$ column elute, Lane 4- SEC fractions. (**B**) An ATPase assay was performed with samples (5 µg) collected at each purification step. Total *E. coli* lipids (10 µg) were added to the SEC fraction. The inactive D373N mutant purified as wild-type MgtA and served as negative control. (**C**) Concentration-dependent activation of ATPase activity by the individual lipid component of *E. coli* inner membrane. Lipids prepared as described in the methods section were added to MgtA at indicated concentrations and phosphate release was measured. (**D**) Comparison of the ATPase activity induced by *E. coli* CL and bovine heart CL. 100% Specific activity represents 13 µM $min^{-1}$ $mg^{-1}$ of MgtA. (**E**). SEC profile of MgtA in the presence of CL and $Mg^{2+}$. MgtA (—); MgtA with 5.0 mM $MgCl_2$(—); MgtA with CL (– –); MgtA with CL and 5.0 mM $MgCl_2$ (...). MgtA and CL were mixed at 1: 3500 molar ratio and incubated on ice for 30 min before SEC. The closed arrow indicates void volume. (**F**) Effect of divalent cations on ATP hydrolysis by MgtA. $Mg^{2+}$(●), $Mn^{2+}$(■), $Ca^{2+}$(▲), $Ni^{2+}$(♦), $Zn^{2+}$(✳), $Co^{2+}$(○). All the conditions with cations have a basal concentration of 3 mM $Mg^{2+}$ and 3 mM ATP. B, C, E, F – Values plotted are mean ± SD (n = 3).

The following source data and figure supplements are available for figure 2:

**Source data 1.** The values represented in the figure are given in excel and the corresponding figure numbers are marked as the sheet name.

**Figure supplement 1.** Purification of MgtA D373N: D373N mutant, which cannot be phosphorylated served as negative control.

**Figure supplement 2.** Effect of DOPE and DOPG on CL activated MgtA.

**Figure supplement 3.** MgtA ATPase activity in the presence of inhibitors.

**Figure supplement 4.** Influence of $Ni^{2+}$ on $Mg^{2+}_{total}$ induced ATPase activity of MgtA.

## ATPase activity of MgtA is restored by cardiolipin and phosphatidyl glycerol

To test whether detergent solubilization and further purification steps had any deleterious effects on MgtA function, we monitored the protein activity using an ATPase assay, which measures phosphate release. A decline in the ATPase activity was observed during purification of wild type (wt) MgtA, with more than 90% loss of the ATPase activity after the final SEC purification (*Figure 2B*). The

gradual decrease in the ATPase activity between each purification step prompted us to test whether the ATPase activity of MgtA was lipid dependent. Addition of total lipid extract from *E. coli* recovered the ATPase activity (*Figure 2B*). We included an inactive mutant (D373N MgtA), which cannot be phosphorylated, as a negative control to rule out the ATPase activity from other endogenous ATPases. The mutant was purified equivalently to wtMgtA (*Figure 2—figure supplement 1*) and did not show any significant ATPase activity when supplemented with *E. coli* lipids.

The phospholipid composition of *E. coli* inner membrane has been reported to consist of ~75% phosphatidyl ethanolamine (PE), ~20% phosphatidyl glycerol (PG) and ~5% CL (*Morein et al., 1996*). To test whether the observed effect of lipids is attributed to a single lipid type, we performed the ATPase assay with increasing amounts of DOPE (1,2-dioleoyl-*sn*-glycero-3-phosphoethanolamine), DOPG (1,2-dioleoyl-*sn*-glycero-3-phospho-(1'-*rac*-glycerol) and CL (isolated from *E. coli*) and found that only anionic lipids such as DOPG and *E. coli* CL restored the ATPase activity in a concentration-dependent manner. DOPG activated MgtA by thirty-fold, while CL by hundred-fold, relative to the lipid-free condition (*Figure 2C*). DOPE increased the ATPase activity only by three folds. Further, a non-native *E. coli* lipid, DOPC (1,2-Dioleoyl-sn-glycero-3-phosphocholine), did not show any measureable effect on the ATPase activity. Apparent $V_{max}$ and $K_m$ calculated for MgtA in the presence of CL are given in *Table 1*.

The maximal ATPase activity observed with total *E. coli* lipid extract (1.7 $\mu M$ $P_i$ $min^{-1}$ $mg^{-1}$) was lower than the maximum ATPase activity observed with pure CL. The possible reasons could either be the low availability of CL in *E. coli* lipid extract or other lipid types like PE and PG competing for the lipid binding site in MgtA. To test the latter possibility, we added increasing amounts of DOPE and DOPG to pre-incubated MgtA-CL mixture and measured ATP hydrolysis. Only a minor decrease in the ATPase activity (<20%) (*Figure 2—figure supplement 2*) was observed, suggesting that other lipids are likely not competing with CL binding. Moreover, the extra DOPG did not show any additive effect over CL induced ATPase activity. The minor decrease in the ATPase activity is likely due to the presence of detergent $C_{12}E_8$ (Octaethylene glycol monododecyl ether) in which the lipids are dissolved (*Figure 2—figure supplement 2*). Therefore, the subsequent enzymatic studies were performed in the presence of *E. coli* CL extract unless stated otherwise.

## The head group of CL plays a major role in MgtA activation

CL isolated from *E. coli* has a diverse fatty acid distribution, with 16:0–18:1 as the most dominant (~35%) and 16:0–17:0Δ as the second most dominant (~17%) species at positions 1 and 2 of *sn*-glycerol 3-phosphate (*Yokota et al., 1980*). To test whether the head group (diacylglycerol phosphate) of CL or the diversity of fatty acid chains plays a prominent role in activating MgtA, we performed ATPase assays in the presence of CL extracted from bovine heart (*Figure 2D*). All the four fatty acid chains of bovine heart CL are almost exclusively linoleate chains (18:2, 18:1, 18:3) (*Shinzawa-Itoh et al., 2007*; *Schlame et al., 1993*). The initial activation profile for both CL extracts are equivalent, which suggest that the head group plays the major role in the activation and not the chemical diversity found within the fatty acid chains in *E. coli* CL. However, the maximal ATPase activity in the presence of bovine heart CL was approximately 30% lower than what was observed with *E. coli* CL. This could mean that the longer linoleate chains from bovine heart CL are suboptimal for effective interaction between MgtA and CL.

**Table 1.** Kinetic property of purified MgtA. The apparent $V_{max}$, $K_m$, and the turnover number ($K_{cat}$) values were determined by least squares fit of the data from *Figure 4C*, as described in materials and methods.

| Parameters | 1 mM ATP | 3 mM ATP | 6 mM ATP |
|---|---|---|---|
| $V_{max}$ ($\mu mol$ $min^{-1}$ $mg^{-1}$) | 14.0 ± 0.2 | 13.7 ± 0.2 | 14.5 ± 0.2 |
| $K_m$ ($\mu M$) | 15 ± 0.6 | 10 ± 0.6 | 10 ± 0.6 |
| $K_{cat}$ ($s^{-1}$) | 23 ± 0.3 | 22 ± 0.8 | 24 ± 0.2 |

## MgtA functions as a monomer and is sensitive to P-type ATPase inhibitors

Whether MgtA acts as a monomer or dimer in the presence of CL will affect further interpretation of the enzymatic studies since CL has been shown to bind and facilitate oligomerization in other proteins like the nitrate reductase (*Arias-Cartin et al., 2011*) and SecYEG (*Gold et al., 2010*). The SEC elution profile shows a monodisperse peak corresponding to the size of a monomer (*Figure 2E*). The SEC profile of MgtA in the presence and absence of CL did not show signs of dimerization of MgtA. However, a significant shift in the elution peak was observed corresponding to a difference of ~50 kDa. This shift likely indicates an increase in lipid-protein-detergent micelle radius caused by binding of CL to MgtA. The presence of $Mg^{2+}$ with CL does not show any further shift, confirming that MgtA function as a monomer in the described assay conditions. The general P-type ATPase inhibitor vanadate as well as specific inhibitors that affect different steps of Post-Albers cycle (*Figure 2—figure supplement 3*) inhibited the MgtA ATPase activity. $AlF_4^-$ and $ADP-AlF_4^-$ showed maximal inhibition, followed by vanadate, whereas $MgF_4^-$ showed only a partial inhibition. Further, the typical F-type ATPase inhibitor azide had no effect on MgtA ATPase activity. Together these results confirmed that MgtA behaves like other thoroughly studied P-type ATPase like SERCA (*Danko et al., 2004*).

## The influence of divalent cations other than $Mg^{2+}$ on MgtA

$Mg^{2+}$ transport by MgtA was studied *in vivo* and reported to show inhibition according to the following order of potency: $Zn^{2+} \geq Mg^{2+} > Ni^{2+} \approx Co^{2+} > Ca^{2+}$ (*Snavely et al., 1989*). Utilizing our ATPase assay, we investigated the effect of divalent ions in vitro in the presence of CL and 3.0 mM $Mg^{2+}$ at pH 7.0. We observed the order of inhibition to be $Zn^{2+} > Co^{2+} > Ni^{2+} > Ca^{2+} > Mn^{2+} > Mg^{2+}$ (*Figure 2F*). Half maximal inhibition of ATPase activity was found to be at < 5 µM for $Zn^{2+}$, 37 µM for $Co^{2+}$, 100 µM for $Ni^{2+}$, 500 µM for $Ca^{2+}$ and 1 mM for $Mn^{2+}$. These values agree reasonably well with the published values (*Snavely et al., 1989*). MgtA has been reported to transport $Ni^{2+}$ as well as $Mg^{2+}$ and $Ni^{2+}$ acting as a competitive inhibitor (*Snavely et al., 1991*). The ATPase assay performed with increasing $Mg^{2+}$ concentration at several fixed $Ni^{2+}$ concentrations, reveal that $Ni^{2+}$-likely acts as a non-competitive inhibitor (*Figure 2—figure supplement 4*) rather than competitive.

## MgtA colocalizes with cardiolipin domains at the poles of *E. coli*

Studies with the fluorescent dye 10-N-Nonyl Acridine Orange (NAO) has shown that CL exists as enriched domains at the poles and division septa in *E. coli* (*Fishov and Woldringh, 1999*). NAO is widely used to image anionic phospholipids in bacteria and displays a specific red-shifted fluorescence emission when bound to CL (*Mileykovskaya et al., 2001*). The online web resource Geno-Base: comprehensive resource database of *Escherichia coli* K-12 (*Otsuka et al., 2015*), reports protein localization images for most *E. coli* proteins. The genes were fused to GFP and overexpressed in the *E. coli* strain K-12. The confocal images of MgtA (gene locus: JW4201) fused to GFP from GenoBase showed clear fluorescence localized at the poles. Since our biochemical data showed that MgtA is activated in the presence of CL, we hypothesized that MgtA could colocalize with the CL rich domains in *E. coli*. To further expand the analysis, we decided to include an N-terminal deletion mutant as well (NΔ31-MgtA). The N-terminal of ecMgtA is rich in positively charged residues and is predicted to harbor intrinsically disordered regions. Similar disordered regions are observed in MgtA and the ortholog MgtB from *S. typhimurium*. The deletion mutant served two purposes; 1) to test whether the positively charged residues were important for the interaction with CL, since activation by lipids binding at the N-terminus had been reported for another P-type ATPase, ATP13A2 of the P5 subfamily (*Faraco et al., 2014*) and 2) to test whether this stretch was important for trafficking to the membrane. We generated constructs of MgtA and NΔ31-MgtA with blue fluorescent protein (BFP) fused at the C-terminus. The confocal images of *E. coli* C43(DE3) overexpressing MgtA-BFP confirmed that it was mainly concentrated at the poles and colocalized with the NAO-stained CL domains (*Figure 3A*). The NΔ31-MgtA-BFP was also found at the poles and colocalized with CL (*Figure 3B*). As a control, the overexpressed BFP alone did not colocalize with the NAO stained CL (*Figure 3C*). Furthermore, no significant difference in the ATPase activity between the purified wt and NΔ31 mutant (*Figure 3D*) was observed. This suggests that unlike ATP13A2, the intrinsic disorder predicted in the N-termini of both MgtA in *E. coli* and MgtA/B in *S. typhimurium* is

not required for activation by lipids and do not play any role in membrane trafficking or targeting to the CL domains (*Figure 3E*).

In addition to the colocalization experiments described above, we also tested localization of the nonphosphorylable mutant MgtA D373N to understand whether the membrane trafficking and co-localization of MgtA with CL at the poles were dependent on the catalytically active MgtA. We observed that the nonphosphorylable mutant D373N fused to green fluorescent protein (GFP) at the C-terminus showed similar localization at the poles (*Figure 3—figure supplement 1*). This leads us to conclude that the active transport of ions through MgtA does not affect trafficking to the membrane or its localization in the membrane.

## Effect of Mg$^{2+}_{free}$ on MgtA ATPase activity

Studying the Mg$^{2+}$ dependency of purified MgtA with an ATPase assay is complicated as it is difficult to decouple the inherent Mg$^{2+}$ dependency of ATP from the vectorial transport of Mg$^{2+}$ by MgtA. To estimate the sensitivity of MgtA to Mg$^{2+}$, we have to make the assumption that Mg$^{2+}$ must be free in solution to be transported through MgtA. The Mg$^{2+}_{free}$ indicates that the Mg$^{2+}$ ions are coordinated only by water and not by ATP, which is also present in the assay mix. It is not

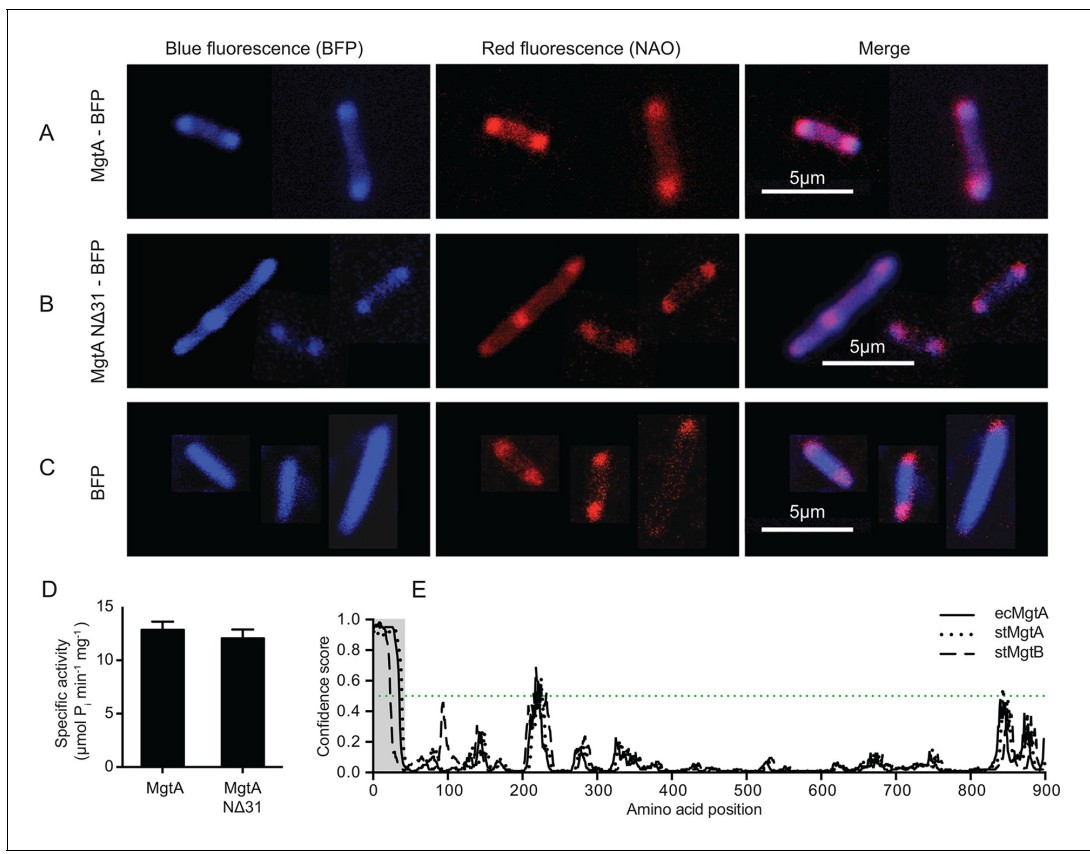

**Figure 3.** Colocalization of MgtA and CL. in *E. coli* C43(DE3). (**A**) Colocalization of MgtA–BFP with CL. (**B**) Colocalization of MgtA-NΔ31–BFP with CL. (**C**) BFP and NAO stained CL. (**D**) ATPase activity of MgtA and MgtA NΔ31 measured in the presence of 125 μM CL. No significant difference in the ATPase activity was observed. (**E**) Disordered regions predicted using DISOPRED3 (*Jones and Cozzetto, 2015*). The polypeptide chain is considered disordered when the prediction is above a confidence score of 0.5. Residues 1–35, marked in grey box shows the disordered region. D, Values plotted are mean ± SD (n = 3).

The following source data and figure supplement are available for figure 3:

**Source data 1.** The values represented in the figure is given in excel and the corresponding figure numbers are marked as the sheet name.

**Figure supplement 1.** Localisation of wtMgtA and the non-phosphorylatable mutant D373N in *E.coli* C43(DE3) cells.

possible to perform an ATPase assay with the concentrations of $Mg^{2+}_{free}$, $ATP_{free}$ and $Mg^{2+}$-ATP acting independent of each other. Thus, it becomes inevitable to obtain the desired information from an assay where one ligand is fixed (e.g. ATP) while varying the concentration of other ligand (e.g. $Mg^{2+}$).

The ATPase activity was measured at three fixed concentrations (1.0, 3.0 and 6.0 mM) of ATP, with increasing concentrations of $Mg^{2+}$ ($Mg^{2+}_{total}$) from 0 to 10.0 mM. This experiment showed steep activation of MgtA ATPase activity at low $Mg^{2+}_{total}$ levels that peak when the molar ATP concentration is equivalent to the molar $Mg^{2+}_{total}$ concentration (*Figure 4A*). A rapid decline in ATPase activity was observed at higher $Mg^{2+}_{total}$ concentrations. For each of the ATP concentrations indicated, the ATPase activity reached equivalent maximum velocity (~14 µmol Pi $min^{-1}$ $mg^{-1}$). However, the activation profile was slightly shifted towards higher $Mg^{2+}_{total}$ as the ATP concentration in assay buffer increased. This shift can be explained by the fact that higher ATP concentrations chelate more $Mg^{2+}$ than lower ATP concentrations, thereby reducing the level of $Mg^{2+}_{free}$ available for MgtA to transport, causing a delay in activation. This is verified by plotting the data from *Figure 4A* against the calculated $Mg^{2+}_{free}$ (Calculated with MAXC, see materials and methods) (*Figure 4B*). We found that the activating concentration of $Mg^{2+}_{free}$ was equivalent (~10.0 µM) for all three fixed ATP concentrations. Further, we observed that the ATPase activity peaked at 250 µM $Mg^{2+}_{free}$ and was inhibited rapidly above 1.0 mM $Mg^{2+}_{free}$. Since the cytoplasmic $Mg^{2+}_{free}$ concentration is ~1 mM (*Silver and Clark, 1971*; *Froschauer et al., 2004*), the inhibition that is observed could occur from the cytoplasmic side of MgtA. In that case, the inhibition can be explained as product inhibition, considering the $Mg^{2+}_{free}$ transported into the cytoplasm to be the product. Together, these results indicate that $Mg^{2+}_{free}$ regulates MgtA ATPase activity when the $Mg^{2+}_{free}$ levels deviate from the 1 mM physiological threshold concentration. We also calculated the apparent $K_m$ for $Mg^{2+}_{free}$ to be between 10–15 µM for the three ATP concentrations tested (*Figure 4C*, *Table 1*) which is only slightly lower than the reported ~29 µM from *in vivo* studies with *S. typhimurium* at 37°C (*Snavely et al., 1989*).

## Effect of increasing ATPconcentration on the ATPase activity of MgtA

Correspondingly in a complementary experiment, the ATPase activity was measured at six fixed $Mg^{2+}_{total}$ concentrations with increasing ATP concentrations from 0–10.0 mM (*Figure 4D*). As observed in *Figure 4A*, the ATPase activity peaked when the molar concentration of ATP was equivalent to the indicated levels of molar $Mg^{2+}_{total}$. However, a slight decrease in the ATPase activity was observed at higher concentrations of ATP for all fixed $Mg^{2+}_{total}$. This could indicate that the $ATP_{free}$ is competing for the $Mg^{2+}$-ATP binding site, thereby reducing the MgtA-$Mg^{2+}$-ATP complex formation. However, interestingly, the activation curves were shifted to higher concentrations of ATP as the $Mg^{2+}_{total}$ in the reaction mix increased. This delay in activation at higher concentrations of fixed $Mg^{2+}_{total}$ once again indicates that high levels of $Mg^{2+}_{free}$ acts as inhibitor as observed in *Figure 4B*, and visualized in *Figure 4—figure supplement 1*. When excess $Mg^{2+}_{free}$ is present at lower ATP concentrations, the ATPase activity is inhibited. As the ATP concentration increases, it chelates more $Mg^{2+}_{free}$ thereby activating MgtA. These results further confirm that MgtA is more sensitive to $Mg^{2+}_{free}$ levels in the environment.

## Effect of pH on MgtA

Protons often act as counter ions for P-type ATPases and as MgtA is classified as a P3–type ATPase it is expected that MgtA may export $H^+$ or another undetermined substance in exchange for $Mg^{2+}$ import (*Faraco et al., 2014*; *Kehres and Maguire, 2002*). To check the influence of pH on MgtA, we tested the MgtA $Mg^{2+}_{total}$ dependency at different pH values (*Figure 4E*). Maximum ATPase activity was observed at pH 7.2. However, at all pH values, a decrease in ATPase activity at higher $Mg^{2+}_{total}$ concentration was observed corresponding to product inhibition as seen in *Figure 4A*. When the ATPase activity was plotted against $Mg^{2+}_{free}$ (*Figure 4F*), it became clear that in the pH range from 6.3 to 7.2, the activation and inhibition profiles were equivalent with only decrease in the maximum ATPase activity when the pH is below 7.0. At pH 7.6 and pH 7.9, the affinity for $Mg^{2+}_{free}$ was reduced, but the inhibition by $Mg^{2+}_{free}$ was equivalent to the other pH ranges. Our data do not show any clear proton dependent $Mg^{2+}$ affinity for MgtA. Considering that the MgtA homolog PH1 from *P. hybrida* did not show proton transport activity (*Faraco et al., 2014*), we have to question

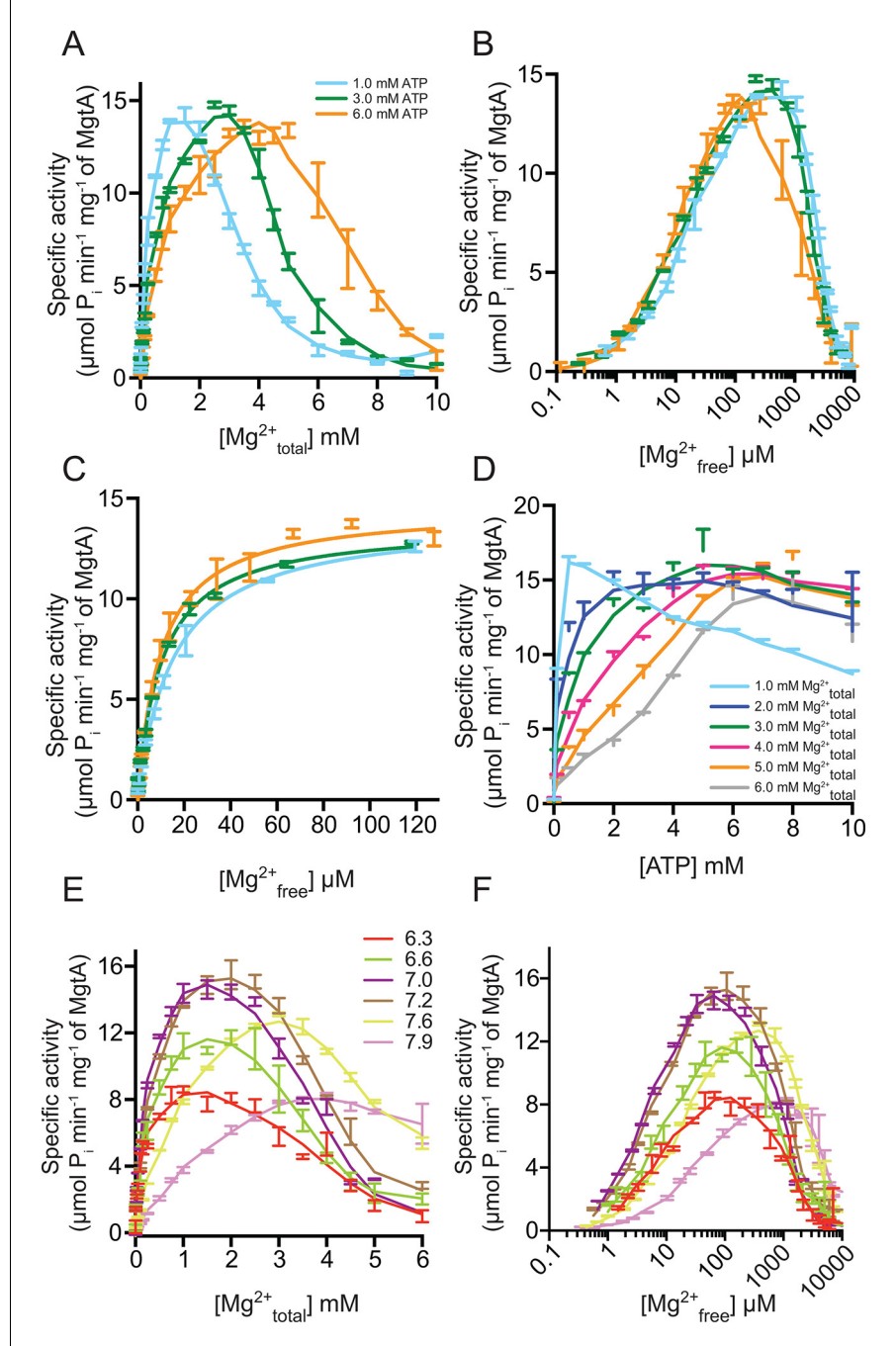

**Figure 4.** $Mg^{2+}$ and pH dependency of MgtA. (**A**) ATP hydrolysis was measured with increasing concentration $Mg^{2+}$ at fixed ATP concentrations. Irrespective of the ATP concentration used, maximum ATPase activity was observed at around 13 $\mu$M $P_i$ $min^{-1}$ $mg^{-1}$ and increasing $Mg^{2+}$ concentration decreases the ATPase activity. (**B**) Data from [A] was plotted against $Mg^{2+}_{free}$ in assay condition determined using MAXC as mentioned in the methods section. (**C**) Data from [B] was plotted in the range from 0 to 120 $\mu$M $Mg^{2+}_{free}$ and used for kinetic calculations summarized in *Table 1*. (**D**) ATP hydrolysis was measured with increasing ATP concentration at various fixed $Mg^{2+}$ concentrations. Since ATP acts as $Mg^{2+}$ chelator, the $Mg^{2+}_{free}$ concentration decreases with increasing ATP concentration. (**E**) ATP hydrolysis measured at various fixed pH conditions while increasing the $Mg^{2+}$ concentration. Bis-tris propane at indicated pH values was used as buffer. (**F**) Data from [E] was plotted against calculated $Mg^{2+}_{free}$. A-F, Values plotted are mean ± SD (n = 3).

The following source data and figure supplement are available for figure 4:

*Figure 4 continued on next page*

*Figure 4 continued*

**Source data 1.** The values represented in the figure is given in excel and the corresponding figure numbers are marked as the sheet name.
**Figure supplement 1.** ATP dependency of MgtA at fixed $Mg^{2+}$ concentration.

---

whether proton counter transport occurs in MgtA. Both MgtA and PH lack the membrane imbedded arginine, which is important for proton translocation as described for the P3A-type ($H^+$) ATPases, Pma1 (Arg 695) in yeast (*Dutra et al., 1998*) and AHA2 (Arg 655) in Arabidopsis (*Buch-Pedersen and Palmgren, 2003*; *Pedersen et al., 2007*). These observations suggest that MgtA may be importing magnesium without the need for proton as counter ion. Since enzymatic activity in general is affected by changes in pH, the pH dependent ATPase activity changes we observe may not be linked to proton translocation. The role of protons as counterions for MgtA therefore awaits further verification.

## Discussion

Over the past two decades, identification of different $Mg^{2+}$ transporters in bacteria and their link to the virulence system PhoP/PhoQ (*Groisman et al., 2013*) has prompted new studies into these transporters. A key $Mg^{2+}$ transporter is MgtA, which is expressed under low $Mg^{2+}$ conditions sensed by the PhoP/PhoQ system. A detailed picture of the transcriptional regulation of *mgtA* is available. However, the in vitro enzymatic characterization of MgtA has been missing and the factors that affect the enzymatic functions are not known. Towards the aim of enzymatic characterization of MgtA, we purified MgtA and demonstrated the tight association of MgtA with CL, and sensitivity of MgtA towards $Mg^{2+}$, providing the first step in understanding the P3B-type ATPases.

In this study, MgtA was overexpressed and purified from *E. coli*. The purification process decreased the ATPase activity of MgtA, indicating the removal of an activating compound or an interacting partner. Addition of total *E. coli* lipids restored the ATPase activity, leading to successful reactivation of purified MgtA. Addition of DOPE, DOPG and CL independently to an ATPase assay revealed that MgtA was only activated by anionic phospholipids. Furthermore, CL showed the strongest activation as compared to DOPG, while DOPE had no effect on ATPase activity. This indicated the importance of specific lipid-protein interactions in the membrane. Our results are the first to link activation by CL to any P-type ATPase. CL has often been linked to membrane proteins connected with the oxidative phosphorylation in mitochondria and membrane proteins involved with lipid metabolism (*Planas-Iglesias et al., 2015*). Lipids exert their effect on membrane proteins either by binding to specific sites or by changing the environment immediate to the protein, thus facilitating oligomerization (*Ernst et al., 2010*). MgtA does not exhibit oligomerization in presence of CL or CL-$Mg^{2+}$ in vitro Thus, CL likely causes a small structural change directly on the protein surface either by binding to a specific site or by facilitating $Mg^{2+}_{free}$ recognition. Our data suggests that the CL binding does not occur at the N-terminus but at an yet unidentified site.

Addition of other lipid types (e.g. DOPE and DOPG) to MgtA pre-incubated with CL did not affect the ATPase activity. This suggests that other lipid types do not interfere with CL binding to MgtA and hints that the head group of the lipids plays an important role in the activation. The preferential selection of CL in the presence of other lipid types hint that MgtA function may be regulated in vivo by changing the local concentration of CL. Since CL have been shown to form domains or rafts at the poles and division septa in *E. coli* (*Renner and Weibel, 2011*), we hypothesized that MgtA could be localized in these CL domains. Our confocal studies indeed showed that MgtA is confined to the poles and colocalized with NAO stained CL. Several publications have validated the use of NAO to stain CL-rich domains (*Mileykovskaya et al., 2001*; *Mileykovskaya and Dowhan, 2000*; *Mileykovskaya and Dowhan, 2009*). However, a recent study questions the specificity of NAO to CL, and shows that NAO stains both anionic phospholipids, PG and CL in the membrane and at the poles of *E. coli* (*Oliver et al., 2014*). Considering the discrepancies in NAO staining, it is not possible to distinguish which anionic lipid among CL and PG would have the most pronounced effect on MgtA in vivoin the *E. coli* inner membrane. It has earlier been reported that expression of

MgtA and consequent removal of $Mg^{2+}$ from the periplasm by MgtA further activated the sensor kinase PhoQ of the two-component system PhoQ/PhoP. This positive feedback regulation increased the level phosphorylated PhoP, which is necessary for the expression of subset of genes expressed under low $Mg^{2+}$ condition *Park and Groisman, 2014*It is interesting to note that PhoP of the two-component system PhoQ/P, which activates MgtA expression, also goes to one pole in *S. typhimurium* when activated by phosphorylation (*Sciara et al., 2008*).Our observation that MgtA is localized in CL rich domains in the poles, thus connect the localization observed for activated PhoP and MgtA. However the exact molecular mechanism behind the similar localization pattern of MgtA-CL and PhoP remains to be determined.

Activation by both anionic lipids has been observed for other membrane proteins, including the anaerobic respiratory complex in *E. coli* (*Arias-Cartin et al., 2011*), the mechanosensitive channel MscL (*Powl et al., 2003*), the protein translocon complex SecYEG (*Gold et al., 2010*) and the cell division proteins MinD and MinE (*Renner and Weibel, 2012*). Thus, it is reasonable to conclude that MgtA is active in an anionic phospholipid environment. This diversity of lipid binding could arise from the structural similarity between PG and CL. In bacteria, CL is synthesized by the fusion of two PG molecules (*Tropp, 1997*). While CL carries two negative charges in the head group, PG carries only one. This difference in the charge could contribute either to efficient binding of lipid or efficient presentation of substrate ($Mg^{2+}$) to MgtA. It has been shown that CL can bind divalent cations like $Mg^{2+}$ and undergo lipid phase changes as a function of $Mg^{2+}$ concentration(*Vail and Stollery, 1979*). It is therefore tempting to speculate that CL could act as a $Mg^{2+}$ reservoir or as a chaperone for easy substrate presentation to MgtA. The $Cu^+$-ATPases from the P1B-type ATPase family are well known to require protein chaperones that are either part of the $Cu^+$-ATPases or expressed as separate proteins in cytoplasm to bind and deliver $Cu^+$ to the $Cu^+$-ATPases (*Arguello et al., 2007*). Our observation with bovine heart CL indicates that the head group of cardiolipin plays the main role in activation of MgtA relative to the composition of fatty acid tails. This, it is intriguing to speculate that the head group of CL could function as the $Mg^{2+}$ chaperone for MgtA and its function could differ depending on the side it associates with MgtA. If the headgroup of CL faces the periplasmic side, then it may act as a chaperon for $Mg^{2+}$, assisting the uptake of $Mg^{2+}$. If the headgroup faces the cytoplasmic side, it may play a role in sensing the $Mg^{2+}$ levels in cytoplasm, thereby regulating MgtA activity. However to further address these questions, a crystal structure of CL bound to MgtA will be necessary as it is not possible to determine the sidedness of the CL headgroup binding site in MgtA in the present study.

Surprisingly, the effect of $Mg^{2+}$ on the ATPase activity of MgtA was more pronounced than expected. MgtA is activated already below 10 μM $Mg^{2+}_{free}$. The activation is followed by a rapid decrease in activity as the $Mg^{2+}_{free}$ concentration rises above 1 mM (*Figure 4B*). All P-type ATPases require $Mg^{2+}$ for their activity. Even though our data does not conclusively establish the transport of $Mg^{2+}$ by MgtA, it shows that MgtA is highly sensitive to the $Mg^{2+}_{free}$ and the level of sensitivity is different from other P-type ATPases. The related $H^+$ ATPase from *Neurospora crassa* also shows decrease in ATPase activity at higher $Mg^{2+}$ concentration (*Brooker and Slayman, 1983*). However, it is not completely inhibited even above 20 mM of $Mg^{2+}_{total}$, whereas MgtA is completely inhibited above 5 mM. MgtA presents a strikingly narrow activity range induced by $Mg^{2+}$ as compared to the $H^+$ ATPase. Overall, we conclude that MgtA is more sensitive to $Mg^{2+}$ concentrations in vitro than any other studied P-type ATPase. This sensitivity towards $Mg^{2+}$ points out that $Mg^{2+}$ is the ion transported through MgtA *in vivo*. The narrow activity range suggests that MgtA could have more $Mg^{2+}$ binding sites other than the transport sites to sense the varying $Mg^{2+}_{free}$ concentrations.

The $Mg^{2+}_{free}$ concentration *of E.* coli and *S. enterica* cytoplasms is found to be ~1 mM (*Silver and Clark, 1971*; *Froschauer et al., 2004*). The observation that MgtA transports $Mg^{2+}$ into the cytoplasm and its ATPase activity decreases above the physiological 1 mM $Mg^{2+}_{free}$, strengthens the notion that the cytoplasmic $Mg^{2+}_{free}$ concentration also regulates MgtA internally. Additionally, it has been reported that increasing cytoplasmic $Mg^{2+}$ concentrations negatively regulates *mgtA*-coding region (*Cromie et al., 2006*). Based on these results we extend the model, initially proposed by Groisman and coworkers (*Cromie et al., 2006*; *Park et al., 2010*) in which PhoP/PhoQ senses low $Mg^{2+}$ in the periplasm and activates *mgtA* transcription. When the cytoplasmic $Mg^{2+}$ falls below a threshold, the riboswitch allows transcription of *mgtA*. Our data adds further evidence to the post translated system, in which MgtA is targeted to the PG-CL domains at the poles and transport $Mg^{2+}$ into the cytoplasm, thereby satisfying the cellular needs. When the $Mg^{2+}$ concentration in the

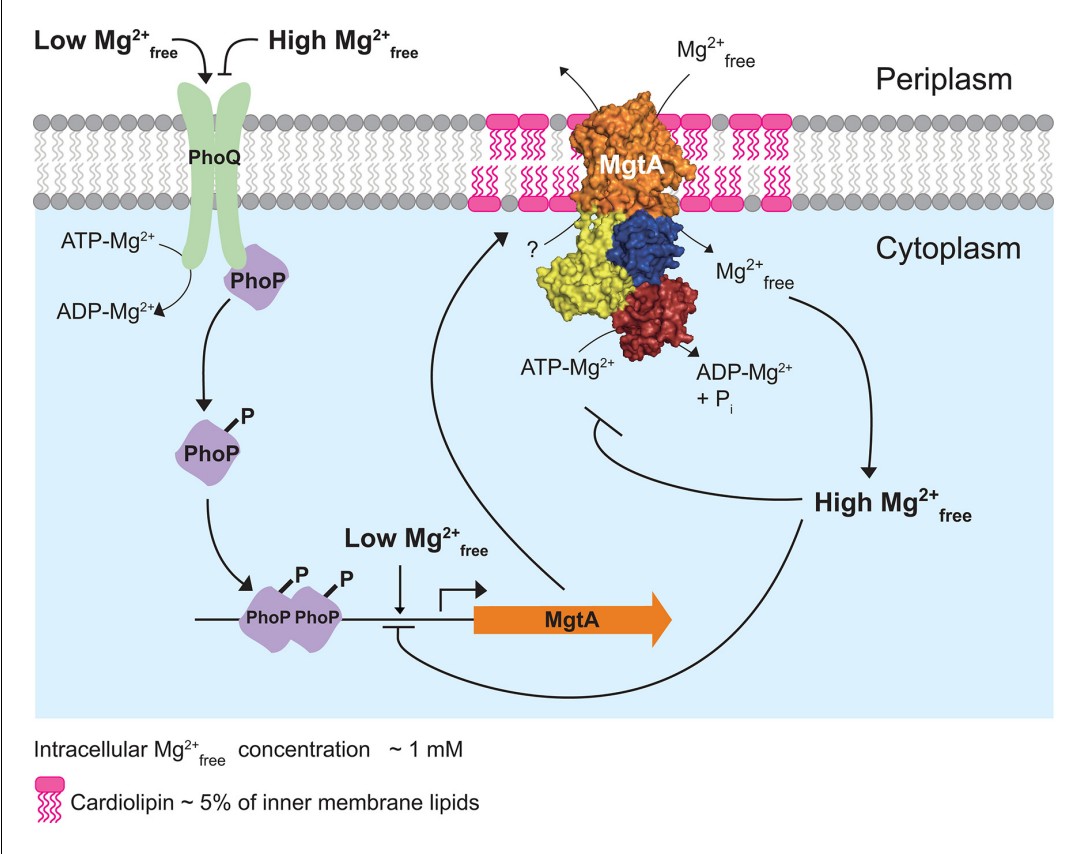

**Figure 5.** Model illustrating the regulation of $Mg^{2+}$ uptake by MgtA. When PhoQ senses low $Mg^{2+}_{free}$ (<50 µM) in periplasm, it phosphorylates PhoP. This activates PhoP and it promotes transcription of the *mgtA* gene. MgtA protein is targeted to CL-rich region in bacterial inner membrane. Association of MgtA with CL is essential for its activity. MgtA imports $Mg^{2+}_{free}$ available in periplasm to the cytoplasm of bacteria, thereby increasing cytoplasmic $Mg^{2+}_{free}$ concentration. When the cytoplasmic $Mg^{2+}_{free}$ concentration reaches a threshold (~1 mM), MgtA is inhibited both at the transcriptional and the post-translational level.

cytoplasm exceeds the threshold, $Mg^{2+}_{free}$ negatively regulates both MgtA in PG-CL domain and *mgtA* coding region (***Figure 5***). In conclusion, our data supports that MgtA acts both as a transporter and as asensor for $Mg^{2+}$.

## Materials and methods

Lipids were purchased from Avanti Polar Lipids, Alabaster, Alabama. Dodecyl- $\beta$-D- Maltopyranoside (DDM) was purchased from Chemical Point UG, Deisenhofen, Germany. Octaethylene glycol mono-dodecyl ether ($C_{12}E_8$) was purchased from Nikko Chemicals, Tokyo, Japan. Adenoside 5'-triphosphate disodium salt hydrate (ATP) was purchased from Sigma-Aldrich, Norway. All other chemicals, including primers were purchased as grade (BioUltra) available from Sigma-Aldrich.

### Cloning and heterologous expression of MgtA

The ORF of *mgtA* was amplified from *E. coli* genomic DNA using primers, 5′ GGCCATGGC TTTTAAAGAAATTTTTACCCGGCTC 3′ and 5′ GGCTCGAGTCATTG CCAGCCGTAACGACGG 3′ purchased from Sigma-Aldrich. The PCR product was inserted into pETM11 vector (EMBL) between *NcoI* and *XhoI* restriction sites. The D373N mutant was created using the Quickchange Lightning mutagenesis kit (Agilent Technologies, Matrix AS, Norway). The sequence of the resulting constructs was verified. Then the MgtA-pETM11 plasmid was transformed into *E. coli* C43 (DE3) cells and plated on Luria Bertani (LB) media with 1.5% Agar and 50 µg/ml kanamycin. Colonies of transformed C43 (DE3) cells were inoculated in LB media containing 50 µg/ml kanamycin and incubated at 37°C

for 16 hr. The following day, 1% of the overnight culture was added to LB media with 50 µg/ml kanamycin and incubated with shaking at 37°C until OD$_{600nm}$ reached ~0.6. Protein expression was induced by adding 1 mM Isopropyl-$\beta$-D-1-thiogalactopyranoside and grown at 18°C with shaking (200 rpm). Cells were harvested after 16 hr by centrifuging at 7000 $\times g$ for 10 min. Cell pellets were stored at -20°C until use.

## Purification

Cell pellet was suspended at 1:10 ratio in buffer A (50 mM HEPES pH 7, 100 mM K$_2$S0$_4$, 10% glycerol, 1 mM Phenylmethylsulfonyl fluoride (PMSF), 5 mM $\beta$-mercaptoethanol) and lysed using High Pressure Homogenizer (C5 model, Avestin, Germany) at 15,000 psi. Unlysed cells and inclusion bodies were removed by centrifugation at 20,000 $\times$ g for 20 min. Resulting supernatant was centrifuged at 100,000 $\times$ g for 2 hr. Mixed membrane pellets were weighed and suspended at 1:10 ratio in buffer B (25 mM HEPES pH7, 100 mM K$_2$S0$_4$, 5% glycerol, 1 mM PMSF, 5 mM $\beta$-mercaptoethanol) using a Dounce homogenizer (Sigma-Aldrich). Membranes were solubilized with 1% DDM for 4 hr. Solubilized membrane suspension was passed through His trap FF (GE Healthcare, Norway). The column was washed with 10 column volumes (CV) of buffer B supplemented with 20 mM Imidazole and ~3 critical micelle concentration (CMC) DDM (0.6 mM). Protein was eluted with buffer B supplemented with 250 mM Imidazole and 3 CMC DDM. Eluted fractions were pooled, concentrated and subjected to size exclusion chromatography (SEC) using a S200 sepharose 16/600 column with Buffer C (25 mM HEPES pH 7, 100 mM K$_2$S0$_4$, 5% glycerol, 1 mM DL-Dithiothreitol (DTT), 3 CMC DDM). Fractions from SEC were pooled, concentrated, flash frozen and stored at -80°C until use. Protein concentration was measured using Bradford method. A description of how the critical steps in the purification was overcome, to yield monodisperse MgtA, was described in detail and published at Bio-protocol (*Subramani and Morth, 2016*).

## Preparation of lipid stocks

Lipid dissolved in chloroform was dried under a nitrogen stream. Dried lipid film was resuspended in MilliQ water at a final concentration of 10 mg/ml with vigorous shaking until no visible aggregates could be detected. 20.0 mM C$_{12}$E$_8$ was added to the lipid suspension and solubilized with shaking for 1 hr at RT. Fresh lipid stocks were prepared prior to enzymatic assays.

## Enzymatic assay

ATP hydrolysis was measured by detecting the released inorganic phosphate following the protocol described by Cariani et, al., (*Cariani et al., 2004*). Reaction buffer contain final concentration of 25.0 mM HEPES pH 7.0, 200 mM KCl, in the presence or absence of 0.25 mM Na$_2$MoO$_4$, 5.0 mM NaN$_3$ and 20.0 mM KNO$_3$. Last three substances were added to inhibit the F-Type ATPase, phosphatase and pyrophosphatase. Unless otherwise indicated, 0.25 µg of MgtA and 116 µM CL were used in the assay. The assay components except ATP-Mg$^{2+}$ were mixed to final volume of 60 µl and preincubated at 37°C for 10 min. Reaction was initiated by addition of 3 mM ATP-Mg$^{2+}$ and further incubated for 10 min at 37°C. The reaction was terminated and phosphate content was detected by adding 75 µl of Solution I (prepared fresh every time by dissolving 0.3 g of ascorbic acid in 3.5 ml water, then adding 5 ml of 1 M HCL, 0.5 ml of 10% ammonium molybdate and 1.5 ml of 20% Sodium Dodecyl Sulphate) and incubated for 10 min on ice. Then, 125 µl of solution II (3.5% sodium citrate and 3.5% bismuth citrate in 1.0 M HCl) was added and incubated at RT for 10 min. Absorbance was measured at 690 nm. The online web service MAXC (maxchelator.stanford.edu) was used to calculate Mg$^{2+}$$_{free}$ levels in the presence of any given concentrations of ATP, pH at 37°C. Apparent V$_{max}$, and K$_m$, were calculated by fitting the curves (specific activity (SA) as a function of Mg$^{2+}$$_{free}$) using Michaelis Menten equation (Y = V$_{max}$*X/K$_m$+X) provided by GraphPad Prism6 (www.graphpad.com). K$_{cat}$ was estimated using the formula K$_{cat}$ = V$_{max}$/[MgtA].

## Sequence analysis

The sequences of the MgtA homologs were identified using the HMMER web server (*Finn et al., 2011*) using default settings with ecMgtA (uniprot code P0ABB8) as search sequence. Selected subsets of MgtA homologs were chosen from each kingdom and aligned using MAFFT (*Katoh et al.,*

*2002*) and edited manually in Jalview version 2.8.2 (*Troshin et al., 2011*), the phylogenetic three was built in Seaview, using the BioNJ algorithm (*Gascuel, 1997*).

## Confocal microscopy

MgtA and NΔ31MgtA were cloned into pRSET-BFP vector between *BamHI* and *NcoI. E. coli* C43 (DE3) cells were transformed with MgtA-pRSET-BFP NΔ31MgtA-pRSET-BFP, MgtAD373N-GFP, empty pRSET-BFP or empty pCFGP vector. Cells were grown overnight in LB medium with 100 μg/ml ampicillin at 37°C, 180 rpm. Fresh overnight cultures were diluted 1:100 in LB medium with 100 μg/ml ampicillin and grown to $OD_{600}$ of 0.6. IPTG at 1 mM was added to induce protein expression.

When indicated, 200 nM nonyl acridine orange (NAO) was simultaneously added to the culture to visualize CL membrane domains. After 3 hr at 37°C, 500 μl of each culture was centrifuged for 5 min at 4000 rpm and 450 μl of the supernatant was removed. The cells were resuspended by pipetting and 2 μl was immobilized on a poly-D-lysine coated glass dish (P35GC-1.5-14-C, MatTek) covered with a 1% agarose-LB pad containing 1 mM IPTG.

Fluorescent images were viewed with a LSM510 (Zeiss) microscope and a 63X oil immersion objective. To minimize the toxicity of high-energy excitation light, the focus was set under phase-contrast conditions and then fluorescence images were captured shortly after the shift to high-energy excitation light. Blue fluorescence from BFP tags (excitation 405 nm, emission 420–480 nm) and red fluorescence from enriched and dimerized NAO in CL domains (excitation 488 nm, emission 600–650 nm) were detected. Fluorescence from GFP was acquired at excitation 488 nm and emission 510–560 nm. Images were obtained and processed using ImageJ software (National Institutes of Health).

## Acknowledgement

The authors would like to thank Robert Brooker, Klaus Fendler and William Stanley for critical reading of the manuscript. Hari Shroff, Prahathees Eswara Moorthy and Damian Dalle Nogare for useful suggestions on how to perform the bacterial imaging. The Norwegian Research Council Funding (FRIMEDBIO) # ES486454 and NCMM core Funding supported this study.

## Additional information

### Funding

| Funder | Grant reference number | Author |
| --- | --- | --- |
| Norges Forskningsråd | ES486454 | Saranya Subramani<br>Harmonie Perdreau-Dahl<br>Jens Preben Morth |

The funders had no role in study design, data collection and interpretation, or the decision to submit the work for publication.

### Author contributions

SS, Acquisition of data, Analysis and interpretation of data, Drafting or revising the article, Contributed unpublished essential data or reagents; HP-D, Acquisition of data, Analysis and interpretation of data, Drafting or revising the article; JPM, Conception and design, Acquisition of data, Analysis and interpretation of data, Drafting or revising the article, Contributed unpublished essential data or reagents

### Author ORCIDs

Jens Preben Morth, http://orcid.org/0000-0003-4077-0192

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
