## [Decision Letter]

Thank you for submitting your work entitled "MgtA is activated by cardiolipin and is highly sensitive to free magnesium in vitro" for consideration by *eLife*. Your article has been favorably evaluated by Michael Marletta (Senior Editor) and two reviewers, one of whom, Eduardo Groisman, is a member of our Board of Reviewing Editors.

The reviewers have discussed the reviews with one another and the Reviewing Editor has drafted this decision to help you prepare a revised submission.

Summary:

The manuscript by Morth and colleagues describes a set of experiments that show several important findings on the biology and regulation of the phylogenetically widespread Mg^2+^ transporter MgtA. The authors demonstrate that: 1) anionic phospholipids are required for the in vitro activity (i.e., ATPase) of the MgtA protein; 2) MgtA co-localizes with anionic phospholipids in the poles of *E. coli*; and 3) the activity of MgtA is highly sensitive to the concentration of free Mg^2+^. This manuscript is well written. Moreover, the authors are cautious by not making more claims than their data and the literature allows. Furthermore, their findings nicely complement and are in agreement with the current knowledge of how expression of the MgtA protein is regulated.

Essential revisions:

The mechanism by which cardiolipin alters MgtA function is unclear. The authors should at least include the autophosphorylation mutant D373N in their imaging analyses presented in Figure 3. Does phosphorylation affect co-localization with CL?

---

## [Author Response]

*The mechanism by which cardiolipin alters MgtA function is unclear. The authors should at least include the autophosphorylation mutant D373N in their imaging analyses presented in Figure 3. Does phosphorylation affect co-localization with CL?*

An additional experiment has been added to shed more light on the molecular interaction between MgtA and CL. To establish whether the head group or the fatty acid chain of CL was most important for MgtA function, we performed an ATPase assay as a function of CL extracted from bovine heart vs. *E. coli* CL. Both CL extractions share the same head group, however vary to a greater extent in their fatty acid composition. This experiment indicates that the head group primarily induced initial activation while the fatty acid chains affected the turnover number. Our observations are described in the section “The head group of CL plays a major role in MgtA activation” as well as in the Discussion. A graph that explains the effect of bovine heart CL has been added as Figure 2 and the P-type ATPase inhibitor profile, which was earlier in Figure 2 has been included as Figure 2—figure supplement 3.

As the reviewers suggested, we tested the localization of the non-phosphorylatable mutant D373N and observed that the mutant D373N showed similar localization profiles as wtMgtA. Both wtMgtA and the D373N mutant were found concentrated at the poles in *E. coli*. We conclude from this that active transport through MgtA does not affect the localization of MgtA. This imaging experiment was performed with MgtA tagged with GFP. The BFP tagged D373N mutant show much lower expression for unknown reasons (Figure 6).

Author response image 1.Overexpression of MgtA D373-BFP, and MgtA D373N-GFP in *E. coli* analysed by Western blotting.Equivalently overexpressed C43(DE3) cells were normalized to OD_600nm_ = 1 and 5µl of the cell lysate was used for western blotting. Blot was developed with Penta-His HRP conjugate antibody (Qiagen). Lane1 – Marker, lane 2 – MgtA D373N BFP, Lane 3 – MgtA D373N GFP. The expression level of the BFP tagged construct is markedly lower than the GFP tagged construct.**DOI:**
http://dx.doi.org/10.7554/eLife.11407.020

Colocalization studies with NAO cannot be performed with the GFP tagged constructs, because, when NAO is bound to the membrane it has an emission maximum at 524 nm, which overlaps with the emission range (510-560 nm) that we used to detect GFP fluorescence. Our results on the GFP tagged constructs are described under ‘MgtA colocalizes with cardiolipin domains at the poles of *E. coli”*’ and Figure 3—figure supplement 1. We conclude from this study that wtMgtA and D373N show equivalent localization at the poles. However, we are not able to confirm that NAO colocalize with the polar distribution of the D373N mutant, as we cannot detect the D373N BFP-tagged construct in the fluorescence microscope.